# Proadrenomedullin in the Management of COVID-19 Critically Ill Patients in Intensive Care Unit: A Systematic Review and Meta-Analysis of Evidence and Uncertainties in Existing Literature

**DOI:** 10.3390/jcm11154543

**Published:** 2022-08-04

**Authors:** Giorgia Montrucchio, Eleonora Balzani, Davide Lombardo, Alice Giaccone, Anna Vaninetti, Giulia D’Antonio, Francesca Rumbolo, Giulio Mengozzi, Luca Brazzi

**Affiliations:** 1Department of Surgical Sciences, University of Turin, 10126 Turin, Italy; 2Department of Anesthesia, Intensive Care and Emergency, Città della Salute e della Scienza Hospital, Corso Dogliotti 14, 10126 Turin, Italy; 3Clinical Biochemistry Laboratory, Città della Salute e della Scienza Hospital, University of Turin, 10126 Turin, Italy

**Keywords:** proadrenomedullin, MR-proADM, SARS-CoV-2, COVID-19, biomarkers, intensive care, endothelitis

## Abstract

Mid-regional proadrenomedullin (MR-proADM) is a new biomarker of endothelial damage and its clinical use is increasing in sepsis and respiratory infections and recently in SARS-CoV-2 infection. We conducted a systematic review and meta-analysis to clarify the use of MR-proADM in severe COVID-19 disease. After Pubmed, Embase, and Scopus search, registries, and gray literature, deduplication, and selection of full-texts, we found 21 studies addressing the use of proadrenomedullin in COVID-19. All the studies were published between 2020 and 2022 from European countries. A total of 9 studies enrolled Intensive Care Unit (ICU) patients, 4 were conducted in the Emergency Department, and 8 had mixed populations. Regarding the ICU critically ill patients, 4 studies evaluating survival as primary outcome were available, of which 3 reported completed data. Combining the selected studies in a meta-analysis, a total of 252 patients were enrolled; of these, 182 were survivors and 70 were non-survivors. At the admission to the ICU, the average MR-proADM level in survivor patients was 1.01 versus 1.64 in non-survivor patients. The mean differences of MR-proADM values in survivors vs. non-survivors was −0.96 (95% CI from −1.26, to −0.65). Test for overall effect: Z = 6.19 (*p* < 0.00001) and heterogeneity was I2 = 0%. MR-proADM ICU admission levels seem to predict mortality among the critical COVID-19 population. Further, prospective studies, focused on critically ill patients and investigating a reliable MR-proADM cut-off, are needed to provide adequate guidance to its use in severe COVID-19.

## 1. Introduction

Proadrenomedullin (pro-ADM) is a 52 multipotent regulatory amino acid peptide expressed in various tissues and organs, upregulated by hypoxia, inflammatory cytokines, bacterial products, and shear stress. Its precursor, mid-regional pro-ADM (MR-proADM), is currently considered an effective biomarker of endothelial damage as its increase in plasma seems to correlate with disease severity [1].

The mechanisms underlying this correlation are poorly defined even if associations with cardiovascular and thromboembolic complications, immunosuppression, and sepsis-like multiorgan dysfunction have been reported [2]. Regarding SARS-CoV-2, an association between MR-proADM levels and virus-induced endothelial damage is assumed. As endothelitis emerges as a prominent feature of the severe COVID-19 disease [3,4], an association between MR-proADM levels and virus-induced endothelial damage has been hypothesized as the pathophysiological mechanisms in COVID-19-induced critical illness seem related to an increased incidence of cardiovascular and thromboembolic complications, immune cell deactivation, and sepsis-like multiple organ failure. A rising number of studies has proposed that virus-induced endothelitis, resulting in impaired vascular blood flow, coagulation, and leakage, may partially explain the development of organ dysfunction and edema [5]. In this sense, since ADM has been shown to play a key role in reducing vascular hyper permeability and promoting endothelial stability and integrity following severe infections [3], MR-proADM might be a potential biomarker of COVID-19 severity and may be able to mimic disease progression, allowing the identification of patients most at risk of developing a severe form of SARS-CoV-2-related illness or multi-organ failure.

If the prognostic role of MR-proADM was demonstrated in the context of pneumonia, sepsis, and septic shock—currently the most studied areas evaluating the predictive capacity of this biomarker [3,6,7]—the pathological mechanism has not been fully clarified; nor is it in the case of severe COVID-19, where most of the studies have a limited size and were designed in the context of a pandemic emergency, with heterogeneity of objectives and study contexts.

To find an answer to uncertainties regarding the role of MR-proADM as a predictive marker of the severity of COVID-19 disease, we systematically present a review of the current literature. The possibility of constructing a meta-analysis capable of establishing the MR-proADM clinical severity cut-off in COVID-19 patients admitted to the Intensive Care Unit (ICU) was subsequently investigated.

## 2. Materials and Methods

This systematic review and meta-analysis followed the Preferred Reporting Items for Systematic Reviews and Meta-Analyses (PRISMA) statement [7]. The protocol was registered prospectively in OSF (DOI 10.17605/OSF.IO/V93EW, link https://osf.io/v93ew/ accessed on 1 May 2022). Since not all studies express values of pro-ADM levels by the same assessment technique, we refer to proADM when including results by all methods and to MR-proADM when levels were determined with the B.R.A.H.M.S. KRYPTOR compact PLUS (Thermo Fisher Scientific, Hennigsdorf, Germany) technique.

### 2.1. Eligibility Criteria

The research was conducted on 25 April 2022; randomized controlled trials (RCTs), non-randomized controlled trials (NRCTs), commentaries, letters, systematic reviews, and meta-analyses published in English and Italian were eligible for inclusion. The meta-analysis was then performed evaluating studies conducted only in the ICU setting to assess if MR-proADM levels may vary in survivors versus non survivors in critically ill patients with severe COVID-19 disease.

### 2.2. Information Sources

This systematic review was performed using Pubmed, Embase, and Scopus databases, and was implemented with the use of registries (clinicaltrials.gov, accessed on 25 April 2022) and gray literature searches.

### 2.3. Search Strategy

To perform the systematic review, the following search strategies were selected:PubMed: “proADM” [All Fields] AND (“COVID-19” [All Fields] OR “COVID-19” [MeSH Terms] OR “COVID-19 vaccines” [All Fields] OR “COVID-19 vaccines” [MeSH Terms] OR “COVID-19 serotherapy” [All Fields] OR “COVID-19 serotherapy” [Supplementary Concept] OR “COVID-19 nucleic acid testing” [All Fields] OR “COVID-19 nucleic acid testing” [MeSH Terms] OR “COVID-19 serological testing” [All Fields] OR “COVID-19 serological testing” [MeSH Terms] OR “COVID-19 testing” [All Fields] OR “COVID-19 testing” [MeSH Terms] OR “SARS-CoV-2” [All Fields] OR “SARS-CoV-2” [MeSH Terms] OR “severe acute respiratory syndrome coronavirus 2” [All Fields] OR “ncov” [All Fields] OR “2019 ncov” [All Fields] OR ((“coronavirus” [MeSH Terms] OR “coronavirus” [All Fields] OR “cov” [All Fields]) AND 1 November 2019:3000/12/31 [Date—Publication])); Embase, Scopus, clinicaltrials.gov, and greylit.org: (‘proadrenomedullin’/exp OR proadrenomedullin) AND (‘coronavirus disease 2019’/exp OR ‘coronavirus disease 2019’).

### 2.4. Selection and Data Collection Process

Search results were exported to EndNote V.X9 (Clarivate Analytics, Philadelphia, PA, USA). Duplicates were automatically removed. The review process was carried out in three steps consisting of title and abstract review process, full-text review process, and risk of bias assessment. For each level, four authors (G.M., E.B., D.L., and A.G.) independently screened the articles with conflicts resolved by a third author (L.B.).

### 2.5. Study Risk of Bias Assessment

To assess the risk of bias, the Risk Of Bias In Non-randomized Studies—of Interventions (ROBINS-i) tool [8] and the Rob 2.0 tool [9] were used for NRCTs and RCTs, respectively. Risk of bias assessment was carried out by four authors (G.M., E.B., D.L., and A.G.) independently; where discrepancies were noticed, a third author (L.B.) was involved to resolve them.

### 2.6. Synthesis Methods

The main outcome was the use of pro-ADM as a prognostic marker in patients with COVID-19. A planned Excel spreadsheet was used to extract data (patient’s characteristics, type of surgery, follow-up periods, outcome measures, and main results). The results of the systematic review were reported in a summary table with the main features described for each study. All eligible studies were evaluated to collect data regarding MR-proADM levels among survivors and non-survivors in ICU population with severe COVID-19 disease. Given that the primary outcome was MR-proADM levels, data presented as median and interquartile range were converted into mean and standard deviation using validated online converters [10]. Estimates of effect were derived from quantitative analysis utilizing Review Manager 5.4 [11]. MR-proADM mean levels and standard deviations were used to evaluate mean differences (MD) with a 95% confidence interval (95% CI). Inverse variance method and random effects were used to assess overall MD. Statistical significance was set at *p* < 0.05. To evaluate the size of the effect of the MD, we considered levels of 0.2, 0.5, and 0.8 as small, medium, and large effects. Heterogeneity was assessed using the I^2^ index, with values of 25%, 50%, and 75% taken to indicate low, moderate, and high levels of heterogeneity, respectively [12].

## 3. Results

### 3.1. Study Selection

The systematic literature search retrieved 65 results in databases and one in registers. A flow chart describing the complete literature search process is reported in Figure 1.

After de-duplication, 26 studies were selected for full-text review. Four papers were then excluded because they did not match the inclusion criteria. After an additional literature check, three papers were included in the systematic review [13,14,15]. A total of 20 articles were submitted to the systematic review.

In order to determine ICU-admitted patients’ pro-ADM cut offs, a new revision of selected studies was made, with the aim to organize data in a meta-analysis. Only 4 studied satisfied meta-analysis inclusion criteria. The reasons for exclusion of the 17 papers were: 12 papers did not consider ICU population, 2 papers evaluated different outcomes (i.e., renal replacement therapy, superinfections) [15,16], 1 analyzed MR-proADM levels among children versus adults patients [17], 1 considered pro-ADM levels with a different technique (bioactive ADM) [18], and 1 was excluded because it presented a population already included in a previously published study [19].

One of the four remaining studies included in the meta-analysis process did not report the standard deviation, and for this reason, was not included in calculation [20]. The meta-analysis was then performed with the three remaining studies.

### 3.2. Systematic Review

#### Study Characteristics

Characteristics of the individual studies are provided in Table 1.

The studies were published between 2020 and 2022. Of the 21 selected articles, 9 enrolled an ICU population, 4 were conducted in an Emergency Department (ED), and 8 had mixed populations. All studies were conducted in European countries except for 2, conducted in Russia: 8 studies were from Italy, 4 from Spain, 2 from the Netherlands, 2 from Germany, 1 from France, 1 from the UK, and 1 from Switzerland. The outcome most frequently considered was mortality. Of the 21 selected articles, 16 agree that the value of proADM predicts mortality or poor outcomes.

The enrollment period, as shown in Table 1, was similar for almost all the studies considered. Other data such as Area Under the Curve (AUC) and proADM considered cut-off are shown in Table 1. All studies considered used as MR-proADM determining levels the B.R.A.H.M.S. KRYPTOR compact PLUS (Thermo Fisher Scientific, Hennigsdorf, Germany) technique, except for one paper [18].

### 3.3. Meta-Analysis

Considering the four studies that were candidates for inclusion in the meta-analysis, one [20] could not be included due to lack of total population number. The other three studies reported MR-proADM admission values in ICU patient populations with critical COVID-19 disease divided by survivors and non-survivors. All studies considered were conducted in 2021. Regarding the country, one was conducted in Spain, one in Italy, and one in the Netherlands.

Among the selected studies, 252 patients were enrolled; of these, 182 were survivors and 70 non-survivors (Figure 2). At the admission to the ICU, the average MR-proADM level in survivor patients was 1.01 versus 1.64 in non-survivor patients. The MD of MR-proADM values in survivors vs. non-survivors was −0.96 (95% CI from −1.26, to −0.65). Test for overall effect: Z = 6.19 (*p* < 0.00001) and heterogeneity was I^2^ = 0% (Figure 2).

All studies were prospective non-randomized clinical trials; therefore, the ROBINS-i tool was applied to assess the risk of bias. The overall risk of bias was low (Appendix A). Publication bias was not tested because of the small number of studies.

## 4. Discussion

This systematic review of the literature highlights the potential role of MR-proADM as a clinical prognostic biomarker in critically ill patients with COVID-19, although a lack of an unequivocal explanation regarding its mechanism of action remains. The growing interest in this promising biomarker and its potential role in the context of COVID-19 pandemic should be underlined. The meta-analysis evaluating only studies conducted in ICU seems to confirm the efficacy of the use of this biomarker, although it deserves further studies to increase the sample size and better define a reliable cut-off. The COVID-19 pandemic has renewed attention to the well-known need for a biomarker capable of differentiating the most critical patients to whom interventions and resources should be targeted. In addition, the characteristics of the new infection—especially at the beginning—highlighted the “weaknesses” of traditional biomarkers, such as procalcitonin and C-reactive protein, but also, d-dimer and cardiac enzymes, which were progressively used as “surrogates” for possible damage mechanisms.

Two and a half years after the onset of SARS-CoV2 pandemic, the importance of the mechanism of endothelial damage at the microvascular level has been widely demonstrated. In this regard, the application of the pro-ADM biomarker in this specific context seemed to be of great interest right from the start, to identify—as early as possible—those patients at greatest risk of poor prognosis. The lack of a univocal explanation for its mechanism of action has not discouraged various authors from considering it in the clinical setting, even if its applications remain varied. Overall, the studies included in our review agree in defining the validity of MR-proADM in the early stages of hospitalization as a prognostic biomarker. Elevated values were found in patients with more severe disease and correlated with statistical significance with patient mortality [35]. This aspect emerged both in the ICU setting and in the ED, opening important perspectives not only in terms of patient allocation but also in terms of possible discharge.

However, although the total number of patients involved in the studies is increasing, there is a huge variation in terms of population, outcome, and methods of assessing MR-proADM (Table 1). The prominent discrepancies that had already emerged in studies on proADM in patients with sepsis and septic shock [36] were further enhanced in the pandemic setting.

The reviewed studies focused on two different populations, represented by ED and ICU patients. Among them, different outcomes were considered, sometimes compromising inter-study comparability (i.e., the use of RRT [16], superinfections [15], children versus adult population [17]).

Another source of dissimilarity is the timing of biomarker testing. Most of the studies evaluated the baseline value of MR-proADM at the patient admission, with a single determination (Table 1). Among the articles considered, only six considered more than a single measurement, but with different intervals (i.e., 3 repeated measurements, daily measurements, etc.) [18,21,23,28,33,34]. However, the role of trend analysis of biomarker values over time is recently emerging in the COVID-19 [28] population, but also in sepsis and septic shock [37].

A clear heterogeneity is also reported on cut-off adopted by different authors, as it was in the more studied context of pneumonia, sepsis, and septic shock [3,6,7], where it seems reasonable to consider a difference within settings (ICU, ED, general wards) and the relative expected severity of patients. As the literature is not consistent in establishing a precise cut-off for increase mortality/severity risk, some authors refer to a value derived from their internal cohort, while others relate to literature-reported previous values.

Considering that establishing a cut-off is one of the most important clinical goals, particularly in the context of a pandemic where a reduction in available resources has been experienced, we propose the use of a meta-analytic approach to determine a clinical severity cut-off derived from available studies on MR-proADM in ICU admitted critically ill COVID-19 patients, excluding all studies involving a mixed population. Our aim was to achieve a possible threshold value for evaluation and access to the critical care area, based on defined endothelial damage and related likely organ failure. Considering cut-off values identified from the available scientific literature (Table 1), MR-proADM cut-off values proposed by Elke et al. among patients with severe sepsis or septic shock (namely 2.75 for low-severity patients and 10.9 nmol/for high-severity patients at baseline) [3] might not represent a useful reference for studies still in progress and/or about to be published. However, those values appear quite in line with the previous cut-offs proposed for respiratory infections, while it appears lower than those identified in sepsis or septic shock [22].

We suggest a cut-off evaluating the values expressed in Table 1 for the ICU population and considering the mean difference of the mean MR-proADM values in the two high- and low-risk populations. It might be emphasized that this meta-analysis cannot be used to propose a MR-pro-ADM cut-off value for disease severity, as this would require an individual-patient meta-analysis followed by ROC curve construction and identification of the pro-ADM value corresponding to the best combination of sensitivity and specificity.

Furthermore, it is essential to note the significant difference between the values proposed in the meta-analysis concerning the ones expressed by Elke et al. (namely 0.96 in our meta-analysis vs. 2.5 in patients with sepsis and 10.9 in patients with septic shock in the manuscript by Elke et al. [3]). The reason for this discrepancy is currently not fully known. Although previous experience on the MR-proADM biomarker is related to sepsis and septic shock, the difference in the cut-offs underlines different physiopathological mechanisms. In septic shock, very high values refer to situations in which significant tissue hypoperfusion is present, with consequent organ failure. Otherwise, in the respiratory failure related to severe COVID-19 disease, the endothelial damage is likely to have a different origin, reflecting the need for specific cut-off values.

As discussed above, the overall number of articles on the subject is still limited. Furthermore, the studies considered show clinical heterogeneity concerning the type of population (ED versus ICU) and its severity, the outcomes, the timing of MR-proADM value(s), the cut-off considered, and the possible role of different confounders.

## 5. Conclusions

Despite the lack of randomized clinical trials and the clinical and methodological reported issues, an increased interest in the use of MRpro-ADM and its physiopathology implications in COVID-19 critically ill patients is emerging. In Europe, the current experience on the use of pro-ADM seems to highlight its validity in the early stages of hospitalization as a prognostic biomarker. High values have been found consistently in patients with more severe disease, both in ICUs and EDs, and correlated with statistical significance with patient mortality. Our meta-analysis confirms a significant difference in MR-proADM values at ICU admission between surviving and non-surviving patients.

Current evidence encourages further prospective and adequate studies on this promising predictive biomarker in the COVID-19 population, providing more specific guidance on its use and specific cut-offs. Other areas to be investigated in the next future are possibly confounding factors and the role of the biomarker trend during the time. 

## Figures and Tables

**Figure 1 jcm-11-04543-f001:**
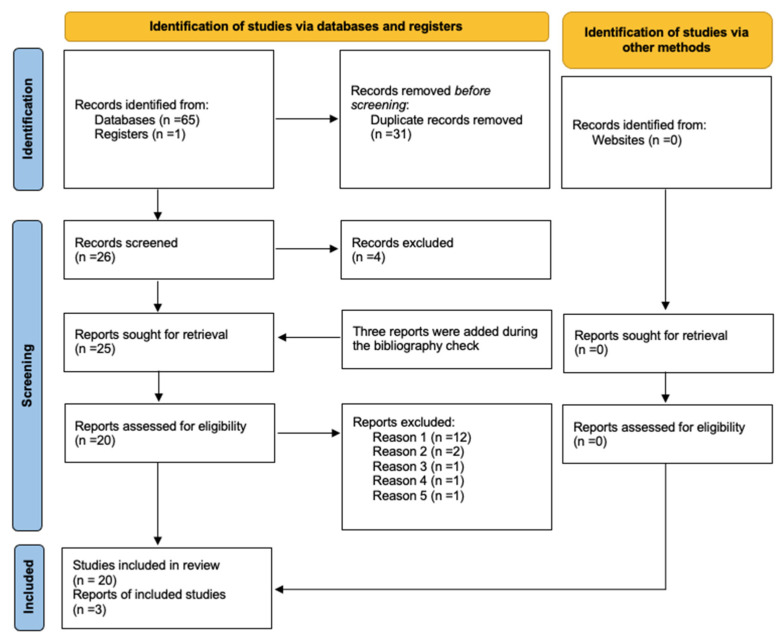
PRISMA flow-diagram. The reasons for exclusion: reason 1: papers which did not consider ICU population; reason 2: papers which evaluated different outcome (i.e., renal replacement therapy, superinfections); reason 3: analyzed MR-proADM levels among children versus adult patients; reason 4: considered pro-ADM levels with a different technique (bioactive ADM); reason 5: presented a population already included in a previously published study.

**Figure 2 jcm-11-04543-f002:**
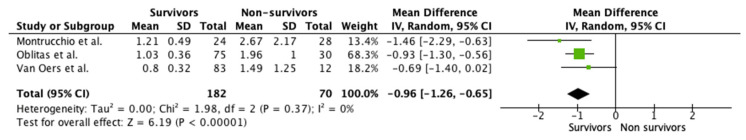
Forest plot of the hypothetical meta-analyzed results [14,28,33]. One of the four studies selected could not be included as it did not report the standard deviation. Analysis conducted with Review manager 5.4 [11].

**Table 1 jcm-11-04543-t001:** Descriptive table of systematic review results, including the 20 full texts analyzed.

Author	Year	Type of Study	Country	Period	Number of Patient	Clinical Setting	Timing	Outcome	Findings	AUC	Cut Off
Benedetti et al. [21]	2021	prospective observational	Italy	March–April 2020	21	IMCU	admission (T0), 24 h (T1), T3 e 5	severe disease	optimal MR-proADM cut-off point was 1.07 nmol/L (sensitivity 91% and specificity 71%) strongest association with 30-days mortality	0.81	1.07 nmol/L
García de Guadiana-Romualdo et al. [22]	2021	prospective observational	Spain	March–April 2021	99	ED	T0	mortality/severe disease progression	highest performance for predicting 90-day mortality low level shows high negative predictive value to rule-out mid-term mortality independent predictor for mid-term mortality; highest prognostic accuracy for short-term mortality	0.871	0.80 nmol/L
Girona-Alarcon et al. [17]	2021	prospective observational cohort	Spain	March–June 2020	20	ICU	hospitalization	pediatric vs. adult population	higher values in children than in adults		
Gregoriano et al. [23]	2021	prospective observational	Switzerland	February–April 2020	89	mixed population	T0, T1, T2, T3	in-hospital mortality	increased 1.5-fold in patients with a fatal outcome safe rule-out of in-hospital mortality in patients with low levels	0.78	0.93 nmol/L
Indirli et al. [24]	2022	retrospective	Italy	March–June 2020	116	IMCU	At admission	in-hospital mortality	with copeptin, predicted in-hospital mortality, occurrence of sepsis or AKI	0.79	>1
Lhote et al. *	2021	prospective multicentric	France	July 2020 to February 2021	170	ICU	T0	SOFA at day 3	insufficient data to confirm proADM validity	NA	NA
Lo Sasso et al. [25]	2021	retrospective observational	Italy	September–October 2020	110	mixed population	hospitalization	Inhospital mortality	good accuracy for predicting mortality	0.95	1.73 nmol/L
Malinina et al. ** [15]	2020	retrospective observational	Russia		37	ICU		Bacterial superinfection	predicts superinfections in patients with SARS-CoV-2 pneumonia		
Mendez et al. [26]	2021	longitudinal	Spain	March–June 2020	210	ED	T0	in-hospital mortality	higher levels in COVID-19 patients associated with poor outcomes a sustained increase is associated with altered DLCO	NA	1.16
Minieri et al. [27]	2021	not specified	Italy	not specified	321	ED	ED-triage	overall in-hospital mortality	key role in the mortality risk stratification at the admission in ED	0.85	1.105
Montrucchio et al. [28]	2021	prospective observational	Italy	March–June 2020	57	ICU	T0–1, T3, T7, T14	ICU mortality—trend	increased plasma levels indicate severity and worse prognosis in CAP, sepsis, ARDS, perioperative care higher values in dying patients predict mortality better than other biomarkers repeated measurement may support a rapid decision-making	0.85	>1.8 nmol/L *
Moore et al. [29]	2022	prospective	UK	April–June 2020	135	ED	at the admission	30-days mortality	predicts 30-day mortality	0.8441	1.54
Oblitas et al. [19]	2021	prospective	Spain	August–November 2020	95	ICU	once within 72 h of ICU admission	30-day mortality and 30-day combined event	predicts 30-day mortality and 30-day poor outcomes	0.73 and 0.72	≥1
Popov et al. [30]	2021	prospective observational	Russia		97	mixed population		mortality	most significant predictor of mortality compared to procalcitonin, saturation and NEWS score.	0.75	0.895 nmol/L
Roedl et al. [16]	2021	observational	Germany	March–September 2020	64	ICU	ICU admission	RRT versus no-RRT	on ICU admission is a strong predictor for RRT early prediction within 24 h after admission	0.69	
Simon et al. [18]	2021	prospective observational	Germany	March–April 2020	53	ICU	Daily, T1–7	ARDS, ECMO, MV, RRT	associated with the severity of ARDS, associated with need for organ support correlation with 28-day mortality		bio-ADM: 70 pg/mL *
Sozio et al. [31]	2021	retrospective	Italy	March–May 2020	111	mixed population	ED admission	severe disease	significantly higher in patients hospitalized with COVID-19 and with negative outcome	0.85	Mortality 0.895 nmol/L
Spoto et al. [32]	2020	prospective observational	Italy	April–June 2020	69	mixed population	hospitalization	endothelial damage, MOF, severe disease	marker of organ damage, disease severity, and mortality values ≥2 nmol/L were associated with a significantly higher mortality risk	0.78	ARDS 3.04; mortality 2 nmol/L
Van Oers et al. [33]	2021	prospective	the Netherlands	March–May 2020	105	ICU	on a daily basis, during the first 7 days	28-day mortalit	with CT-proET-1 is able to identify patients with worst outcome significantly higher levels of MR-proADM and CT-proET-1 in non-survivors persisted over time	0.84	1.57
Zaninotto et al. [34]	2021	retrospective	Italy	November	135	mixed population	7 days	clinical outcomes	additional clinical value in stratifying risk and establishing the prognosis	0.900	1.50

*List of abbreviations*: Area Under the Curve, AUC; Emergency Department, ED; Intensive Care Unit, ICU; Intermediate Care Unit, IMCU; T: time express in days; Multiorgan Failure, MOF; Acute Respiratory Distress Syndrome, ARDS; Extracorporeal Membrane Oxygenation, ECMO; Diffusing capacity for carbon monoxide, DLCO; Mechanical Ventilation, MV; Renal Replacement Therapy, RRT; C-terminal proendothelin-1, CT-proET-1; MR-proadrenomedullin, MR-proADM; Sequential Organ Failure Assessment, SOFA. ** only abstract available. ** full-text article provided by the corresponding author.*

## Data Availability

The datasets used and analyzed during the current meta-analysis are available from the corresponding author upon reasonable request.

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
