# Peer review of "Proadrenomedullin in the Management of COVID-19 Critically Ill Patients in Intensive Care Unit: A Systematic Review and Meta-Analysis of Evidence and Uncertainties in Existing Literature"

_jcm, 2022, doi:10.3390/jcm11154543_

Round 1

Reviewer 1 Report

Thank you for the opportunity to review this manuscript intitled “Pro-Adrenomedullin in the Management of COVID-19 critically ill patients in Intensive Care Unit: a Systematic Review and meta-analysis of Evidence and Uncertainties in Existing Literature”.

In this paper, the authors conducted a systematic review and meta-analysis study on the prognostic value of Pro-Adrenomedullin in COVID-19 critically ill patients.

The main findings are:

1)    Twenty-one studies were published between 2020 and 2022 from European Countries, but only 4 studies evaluating survival as primary outcome in ICU patients were available (3 reporting completed data). A total of 252 patients were enrolled (182 survivors and 70 non-survivors).

2)     At ICU admission, the mean differences of MRproADM values in survivors versus non-survivors was -0.96; 95% CI (-1.26; -0.65). 

3)    Test for overall effect was Z = 6.19 (P < 0.00001) and Heterogeneity was I2=0%. 

4)    The authors concluded MR-proADM ICU admission levels seem to predict mortality among the critical COVID-19 population. 

The authors should be commended on their efforts to provide further insights about the prognostic value on a novel biomarker in COVID-19 patients. They also underlined herein the need for predictive studies to assess the prognostication value of MR-proADM in ICU COVID-19 patients.

This paper is interesting, findings are well described and the manuscript is appropriately organized.

I have only one minor comment: it would be useful to add a figure or a scheme to better explain the difference between MR-Pro-ADM, bioactive ADM and pro-ADM. In the same line, the authors have chosen to focus on MR-Pro-ADM, they have to change the words “pro-ADM” in the conclusions to “MR-Pro-ADM”. 

One more time, thank you for the opportunity to review your manuscript.

Author Response

We appreciate the Reviewer’s insightful comment. We followed the guidance and have created a supplementary table as follows:

Full name

Short name

Biochemical characteristics

Functional characteristics

Biomarker pitfall

Citations

precursor pro-hormone of ADM–mid regional pro-adrenomedullin

MR-proADM

longer half-life than ADM

prognostic value in disease

no known action by itself

Marino R. et al. 10.1186/cc13731.

bioactive adrenomedullin

bioADM

active, short-lived circulating peptide

associated with severe organ dysfunction and an elevated mortality risk

Less used in the literature

Laterre PF et al. 10.1007/s00134-021-06537-5

Plasma adrenomedullin

ADM

vasodilatory peptide

prognostic value in disease

short half-life

Marino R. et al. 10.1186/cc13731.

Reviewer 2 Report

Thank you for reviewing the paper from Montrucchio et al. entitled `pro-adrenomedullin in the management of COVID-19 critically patients in intensive care unit: a systemic review and meta-analysis of evidence and uncertainties in exiting literature. I read the paper with many interests. The covid-19 pandemic, its diagnosis, treatment, epidemiological impact was the leading topic worldwide. The paper from Montrucchio now addressing the inflammatory marker pro-ADM and its impact as prognostic marker and outcome in critical ill patient. Therefore, the authors prepared a systemic review and meta-analysis based on the latest literature. To summarize their findings pro-ADM admission levels might be a prognostic marker, but the author critical reviewed the results in context of the available literature and published studies with many uncertainties and heterogeneities with respect to pro-ADM as outcome parameter. There is an urgent need to evaluate inflammatory markers not only for its prognostic impact in COVID-19. Many aspects have to consider monitoring patient severity, derive treatment strategies, and predict outcome respectively. Here, the authors presented an interesting review and meta-analysis, but a few points have to be addressed.

1 1)      The word `management´ in the title might be little misleading. The most studies no therapeutic changes resulted depending on pro-ADM cut off values.

2 2)      The introduction capital could offer little bit more details regarding aspects to endothelial damage in association to pro-ADM and the context of the literature to other markers should be discussed more detailed.

3 3)      Please check the patient n of the part entitled `included´ in figure 1.

4 4)      Please, rearrange table 1 to get more comfortable the main information by reading it. Now its little confusing, especially column entitled `findings´. It would be helpful to present each definition of the term `timing´ for each study, now it is not really clear for all of it. And please, proof the presented patient numbers, AUC and cut off values. There is discrepancy what Garcia de Guadiana-Romualdo et al. presented in the original study. Lhoto et al. is missing? I could not find the study. The provided literature is a conference abstract and was during the review not available to proof the data in detail. Malinina et al. and Popov et al. also were not available online? Please proof the provided AUC of Zaninotto et al., is it the right one?

5 5)      The meta-analysis has critical to discuss. The pro-ADM might impact patient outcome. But the small number of studies, the heterogeneities of the cohorts, the missing impact of pro-ADM, also in comparison to established markers or clinical scores, the impact of co-infections, and the different cut off values are critical points.

6 6)      The refered supplementary figure 1 is missing or was not provided at the online reviewer platform.

Author Response

We thank the reviewer no 2 for his valuable work on revising our manuscript. We answered each point raised, with a point-by-point replay, as follows:

  • The word `management´ in the title might be little misleading. The most studies no therapeutic changes resulted depending on pro-ADM cut off values.

****Response: We appreciate the Reviewer’s question and the opportunity to clarify. However, in the concept of “management”, we clinically include not only a treatment pathway, but a more general clinical and logistical management of patients. We believe that for everything expressed in the draft and in the reference literature, it may be appropriate to define this concept as "management".

  • The introduction capital could offer little bit more details regarding aspects to endothelial damage in association to pro-ADM and the context of the literature to other markers should be discussed more detailed.

****Response: We thank the Reviewer for the excellent suggestion. As required, a further sentence explaining the pathophysiological mechanism has been added, with the relative further bibliographic references.

The pathophysiological mechanisms in COVID-19 induced critical illness seems related to a particular process of endothelial damage – named “endotheliitis” - causing an increased incidence of cardiovascular and thromboembolic complications, immune cell deactivation and sepsis-like multiple organ failure. An increasing number of studies has proposed that virus-induced endothelial dysfunction and damage, resulting in impaired vascular blood flow, coagulation and leakage, may partially explain the development of organ dysfunction and oedema. In this sense, since adrenomedullin (ADM) has been shown to play a key role in reducing vascular hyper permeability and promoting endothelial stability and integrity following severe infection, MR-proADM might be a potential biomarker of COVID-19 severity (…)

  • Please check the patient n of the part entitled `included´ in figure 1.

****Response: We apologize for the lack of clarity. As it is written in the figure, 20 studies were included in the review, while only three of them were elegible for the meta-analyisis. We corrected in the paragraph ‘Study selection’ the number of the studies selected for the systematic review.

  • Please, rearrange table 1 to get more comfortable the main information by reading it. Now its little confusing, especially column entitled `findings´. It would be helpful to present each definition of the term `timing´ for each study, now it is not really clear for all of it. And please, proof the presented patient numbers, AUC and cut off values. There is discrepancy what Garcia de Guadiana-Romualdo et al. presented in the original study. Lhoto et al. is missing? I could not find the study. The provided literature is a conference abstract and was during the review not available to proof the data in detail. Malinina et al. and Popov et al. also were not available online? Please proof the provided AUC of Zaninotto et al., is it the right one?

****Response: Thank you for your comment. We carefully revised Table 1 and all the data, as required. The presented patient number, AUC and cut off values are all in line with the cited studies. Also, bibliography references have been re-checked and we confirm the reported data. In particular:

Garcia de Guardiana-Romualdo: we corrected the values.

Lhote: the full text articole was not available. We refer to abstract data, included as required in the PRISMA methods. For clarity, we added in the footnotes *only abstract available.

Popov: the full text is available according to personal research academic resources.

Malinina: the full text (not in English (Russian); only abstract in English was provided by email by the corresponding Author.

Zaninotto: we corrected the AUC value.

Concerning the timing, we added a definition in the footnotes the definition.

Concerning the column “findings”: in our opinion, this column provides a summary of each study specific findings. Language is synthetic according to editorial need for a Table.

  • The meta-analysis has critical to discuss. The pro-ADM might impact patient outcome. But the small number of studies, the heterogeneities of the cohorts, the missing impact of pro-ADM, also in comparison to established markers or clinical scores, the impact of co-infections, and the different cut off values are critical points.

****Response: We totally agree with this point. In fact, in the discussion we wrote: The meta-analysis evaluating only studies conducted in ICU seems to confirm the efficacy of the use of this biomarker, although it deserves further studies to increase the sample size and better define an adequate cut-off.

(…)

Considering that establishing a cut-off is one of the most important clinical goals, particularly in the context of a pandemic, where a reduction in available resources has been experienced, we propose the use of a meta-analytic approach to determine a clinical severity cut-off derived from available studies on MR-proADM in ICU admitted critically ill COVID-19 patients, excluding all studies involving a mixed population. The aim of our purpose was to achieve a possible threshold value for evaluation and access to the critical care area, based on defined endothelial damage and related likely organ failure. Considering cut-offs values identified from the available scientific literature (Table 1), MR-proADM cut-offs values proposed by Elke et al. among high-severity population (namely 2.75 and 10.9 nmol/) 3 might not represent a useful reference for studies still in progress and/or about to be published. However, those values appear quite in line with the precedents cut-offs proposed for respiratory infections, while it appears lower than those identified in sepsis or septic shock 22.

As suggested, we added the possible role of confounders: As discussed above, the overall number of articles on the subject is still limited. Furthermore, the studies considered show clinical heterogeneity concerning the type of population (ED versus ICU) and its severity, the outcomes, the timing of MR-proADM value(s), the cut-off considered, the possible role of different confounders.

Moreover, in the conclusion, we highlighted all the points raised by Rev. 2.

  • The referred supplementary figure 1 is missing or was not provided at the online reviewer platform.

****Response: We apologize for the misunderstanding; we have not uploaded supplementary files in the first version of the manuscript. Now you can find it attached.

Round 2

Reviewer 2 Report

Thank you for the revised version. After revsion nothing to add in the present way.

Author Response

Thank you as well to allow us to improve our work!

This manuscript is a resubmission of an earlier submission. The following is a list of the peer review reports and author responses from that submission.